# Using text data instead of SIC codes to tag innovative firms and classify industrial activities

**Alessandro Marra**[1,2]*, **Cristiano Baldassari**[2,3]

**1** Department of Economics, University d'Annunzio of Chieti and Pescara, Pescara, Italy, **2** Explo, academic spinoff, University d'Annunzio of Chieti and Pescara, Pescara, Italy, **3** Department of Neuroscience, University d'Annunzio of Chieti and Pescara, Chieti, Italy

* alessandro.marra@unich.it

## Abstract

The paper uses text mining and semantic algorithms to tag innovative firms and offer an alternative perspective to classify industrial activities. Instead of referring to firms' standard industrial classification codes, we gather information from companies' websites and corporate purposes, extract keywords and generate tags concerning firms' activities, specializations, and competences. Evidence is interesting because allows us to understand 'what firms do' in a more penetrating and updated way than referring to standard industrial classification codes. Moreover, through matching firms' keywords, we can explore the degree of closeness between the firms under observation, a measure by which researchers can derive industrial proximity. The analysis can provide policymakers with a detailed and comprehensive picture of the innovative trajectories underlying the industrial structure in a geographic area.

## Introduction

Innovative firms can seize opportunities created through technological progress, and generate demand for skilled labour, higher wages, and productivity gains [1]. Given the role they play, it is essential to correctly enucleate innovative firms and their industrial activities.

An increasing number of scholars argue that it is not sufficient to refer to the standard industrial classification (SIC) codes to realize firms' activities and appreciate their degree of innovativeness [2, 3]. SIC codes are a consolidated taxonomy and are currently used for the statistical surveys of economic activities, but they show many limitations as widely discussed in the literature.

The information attached to the code itself is scarce: this is limited, in fact, to a very short textual definition (e.g., in Europe NACE code M72.1.1 is equivalent to 'Research and experimental development on biotechnology'). The codes are limited with respect to the existing variety of business activities. Although the codes in the current SIC are more than a thousand, they are not (and probably never will be) enough to describe the abundance of firms' activities. Such codes, which are obsolete and not up to date with technological and market trends, go

**Data Availability Statement:** We made the minimal data set related to the values used to build analysis and graphs publicly available without restrictions on the web repository GitHub,

accessible at the following link: https://github.com/cbaldassari/2022-plosone.

**Funding:** Fondirigenti G. Taliercio (Interprofessional fund for continuous training of managers promoted by Confindustria and Federmanager; www.fondirigenti.it) provided funding for the Strategic Initiative 'Modelling of an Observatory to monitor the innovation ecosystem in the Chieti and Pescara area and mapping of managerial skills with a high innovation rate', grant number: CIG 8188368708. The funder had no role in study design, data collection and analysis, decision to publish, or preparation of the manuscript. The specific roles of the authors AM and CB are articulated in the 'author contribution' section.

**Competing interests:** The authors AM and CB founded the academic spinoff Explo. Explo is specialized on text analytics. This does not alter our adherence to PLOS ONE policies on sharing materials and data. Explo does not have any patent, patent application, or product in development or for market related to the research article.

'tight' to innovative firms. Sometimes firms opt for residual and broader codes. For example, in Europe the NACE code M74.9 –'Other professional, scientific and technical activities': these definitions are too broad and vague to be informative on firms' activities. Finally, we know that firms innovate, transform, and renew continuously: the code chosen yesterday can only partially reflect what the firm does today or will do tomorrow [4].

Instead of referring to SIC codes, we gather information from firms' websites and corporate purposes, extract keywords and generate tags concerning firms' activities, specializations, and competencies. Our methodology assumes that firms use the same (or similar) words to identify and describe the same (or similar) activities, regardless of the SIC codes chosen during the registration phase.

Using text data as input to research is not new in the economic literature [5]. The information encoded in digital texts represents a useful complement to more standard and structured data, and this is testified by the remarkable growth of economic research in recent years that uses texts as data. In finance, for example, text from newspapers and social media is used to forecast stock price movements [6]. In microeconomics, text from advertisements and product reviews allows for the study of consumer decision drivers [7]. In industrial economics, text describing products is used to propose alternative industry classifications [8].

Analysing textual data is a challenging activity when the information is embedded 'somewhere' within large masses of unstructured data. Using text data means starting a step backwards. Once the suitable data source has been identified, the text needs to be manipulated and elaborated into meaningful patterns of understanding and insightful perspectives.

Our paper uses text mining and semantic algorithms to tag innovative firms and offer an alternative perspective to classify industrial activities. Evidence is interesting because allows us to understand 'what firms do' in a more penetrating and updated way than referring to SIC codes. Moreover, by matching firms' keywords, we can explore the degree of interconnection between the firms under observation, a measure by which researchers can derive industrial proximity. The paper is organized as follows. The next sub-section presents the relevant literature: starting from the limitations of the SIC codes, the focus is on new approaches and methods based on text data to describe the content of firms' innovative activities. Section Data illustrates the dataset: the analysis has been carried out on a sample of 583 innovative firms active in Chieti and Pescara, in Abruzzo, Italy. The choice of such a geographic area is motivated by the presence of a large and diversified production system, characterised by numerous and remarkable innovative firms. The next Section illustrates the methodology. A synthetic but comprehensive description of the methodological and operating steps is provided, sacrificing some technical passages and an in-depth discussion of the text mining and semantic algorithms. Even if these algorithms constitute the central part of the investigation, an exhaustive discussion does not seem to be justified for two orders of reason. First, in computational science such algorithms have become a standard with countless applications in many fields of study. Secondly, the algorithms adopted do not show any special advantage compared to others, or at least this is not what we intend to argue in this paper. Section Results emphasizes the abundance of the information that can be attained from the collection and elaboration of text data. Also, a discussion is proposed of the advantages that derive from organizing the labels in different levels or categories of interest and of the novelty of the different views that can be proposed to appreciate the degree of proximity between firms. In fact, once the keywords have been created and sorted into categories, the analysis turns on the resulting matrices of adjacencies by which to bring close one firm to each other based on the number of co-occurrences of categories and keywords between them. For the sake of simplicity, instead of focusing on networks of firms (583 nodes), which would be too large to investigate in detail in this paper, we aggregate firms by sectors (36 nodes) approaching them based on firms' common

specializations and competencies. The last Section concludes, emphasizing limitations and highlighting policy implications.

## Background

In the modern world, it is crucial to identify correctly and readily innovative firms and classify their industrial activities. However, this information is not easy to elaborate on. It is well known that the starting point for understanding what firms do is the SIC system. Using SIC codes is a common practice, even though SIC codes can be uninformative or misleading. The SIC system has many limitations. The descriptions adopted are too concise. The codes, though numerous, are insufficient to represent the variety of existing economic activities [9]. The SIC codes are often obsolete and out of step with technological evolution [3]. Sometimes, to avoid being 'confined' to specific activities or areas, firms opt for broader residual codes. Finally, firms are constantly changing and renewing their businesses: the code chosen yesterday may only partially reflect what the firm does today [10].

In addition to the criticalities mentioned above, there is a legitimate question. In the era of big data, in which firms make available a lot of information that can be collected and processed to classify in a detailed and updated way their industrial activities, why not attempt to make some use of it? Many recent contributions propose original methods starting from text data to classify firms and industries. Below we propose a brief review of the key literature.

[2] develop a sector-product approach and employ text mining to enrich the description of the firm's activities in the ICT and digital sector in the United Kingdom. The authors use raw text data and contextual information gathered from websites and news feeds. Interestingly, they affirm that using text mining might provide further detail over SIC codes, which tends to lag far behind technological evolution [11]. propose the web and new methods of data extraction to derive metadata useful for the industry classification by looking at a regional case in the Northeast of England. The exercise proposed by the authors is a tool to identify specific aggregates of industrial activities in a geographic area. The discussion starts by highlighting the limitations of SIC codes and is followed by the proposed methodology, based on web-based data collection, pre-processing and analysis, and reporting of clusters [12]. conduct a bottom-up study with the aim of overcoming the limitations of industry classifications to study the composition of the economy. Applying machine learning and graph theory techniques, the authors analyze company descriptions extracted from company websites and generate alternative taxonomies on the basis of which they define industries as 'communities within word networks'. Sometimes the researchers' intentions do not stop at describing firms' activities in a new and original way and go further to explore their innovative activities [13]. formulate a novel approach to estimate firms' innovation activity based on texts on corporate websites. They use automated web-scraper to harvest text from websites, then extract semantic topics in a self-learning, generative topic-modelling approach, and analyze these topics using a neural network method to assess each firm's level of innovation.

Increasingly, existing databases are being used to exploit the huge amount of structured and constantly updated data. They range from large proprietary databases to open repositories: in the former textual data, such as information on the companies' corporate mission, registered patents and economic and financial news published in the press, are largely available, while in the latter, there are descriptive summaries of firms' activities and products/ services sorted by keywords [14]. analyze the unstructured texts that describe firms' businesses using the statistical learning technique of topic modeling and construct a proximity measure based on the Latent Dirichlet Allocation algorithm, by which represent each firm's textual description as a probabilistic distribution over a set of underlying topics [15]. perform text mining on

Crunchbase to work on green-tech firms in San Francisco, New York, and London. Using metadata the authors classify firms' industrial activities and underlying specializations, building links for technological and market complementarities, identifying specific firms' aggregates and emerging industrial clusters [16]. measure innovative digital firms' specializations and competencies based on the degree of digital technologies in the products and services supplied. The method allows to overcome the limitations of defining industrial specializations in digital industries through SIC codes and capture innovative firms' specializations at the metropolitan level [17]. propose a classification of specializations along the automotive supply chain in Italy based on the analysis of the descriptive texts of the activities provided in the process of registering with the Chamber of Commerce. The authors implement a multidimensional analysis of words to identify clusters of specializations, and a similarity analysis of words to provide indications on clustering of specializations as they are described by companies.

Sometimes it has been useful to look at alternative information sources than SIC codes to classify economic activities. In this vein are the studies by [8, 18, 19]. [8] collect business descriptions from thousands of firm 10-K product descriptions using web crawling algorithms and process the text to propose new industry classifications. The authors can study how firms differ from their competitors using new time-varying measures of product similarity, which allows for the generation of a new set of industries in which firms can have their own distinct set of competitors.

Once firms' activities have been classified, scholars start to address further research questions. One of these concerns the conditions and modalities by which firms exchange knowledge, promoting its diffusion and re-generation. Such an exchange becomes a knowledge flow, something not easily traceable. In this case, economists are inclined to use proxies, assuming that these flows have a higher chance to occur between firms that are 'closer' in the industry space. In the first place, economists use the SIC system to assume some exchange of knowledge if two firms share the same two-, three-, four-, or five-digit codes [20, 21]. However, such measures are still discrete, and the level of granularity is constrained by the adopted classification system. Moreover, whether such measures are indicative of industrial proximity can be questioned [22]. The most recent attempts to define and classify industrial activities based on alternative data sources all go hand in hand with the next step of operationalizing proximity between firms. Accordingly, albeit not in depth, this article proposes to explore industrial proximity based on the resulting matrices of adjacencies between firms.

## Data

The dataset used for our analysis merges information from different databases, including startup.registroimprese.it by Unioncamere and Analisi Informatizzata delle Aziende (Aida) by Bureau Van Dijk: the former is the official database of the Chambers of Commerce that collects Italian startups and innovative SMEs; the latter includes comprehensive information concerning corporate purposes and financial indicators on companies in Italy.

The initial perimeter includes several hundred firms based in the provinces of Chieti and Pescara, in Abruzzo. Abruzzo has a production system of quality and excellence, large and diversified. Abruzzo is seventh in Italy for industrial expertise and for the impact of exports on GDP, sixth for trade surplus and second for exchange value [23]. The province of Chieti is specialized in the automotive sector, with large firms operating in the production of vehicles and components engineering. In the province of Pescara there is one of the most competitive supply chains in sanitary products in paper and cotton, from machinery to the production of materials, up to packaging [23]. In the provinces, there are also the paper and paperboard industry, activities related to the mining industry, professional, scientific, and technical

activities, and advertising agencies. Other important industries are pharmaceuticals and related industries [23].

To generate and assign keywords to firms, we use different sources of text data such as the company website, the corporate purpose, and synthetic descriptions of the firm's economic activity. On the data sources, we emphasize the importance of the company website that well explains firms' innovative activities [24]. The corporate purpose does not help when too vaguely defines firms' core businesses, which might happen when there is an interest to 'leave open' possible future paths of development and activities. In such a case, we privilege the website, the more updated picture of the business activity. Otherwise, the corporate purpose might be very informative: innovative start-ups and SMEs confirm this. This is explained by the strict assessment carried out in Italy by the local Chambers of Commerce before registration. In all those cases in which the corporate purpose is formulated in a clear and defined way, it represents a useful source of information, which enriches and complements the website.

We employed a specific procedure to enucleate a limited number of innovative firms over which to perform our text mining and semantic algorithms. Even though such a procedure is out of the scope of the present paper, we sum up the adopted reasoning below. The presence of specific keywords (such as 'innovation' and 'technology' in their different declinations) in the text describing the firms helped us to capture some 'essential traits' of innovation that, together with the existence of further terms on the firms' positioning on the market and/ or their ability to export (e.g., 'leadership', 'export' and 'international' in their different declinations), allowed to narrow the perimeter of the analysis. Also, a scoring of confidence and presence attached to a list of keywords related to recent technological advancements (e.g., 'Digital technologies', 'Artificial intelligence', 'Industrial automation', 'Robotics', 'Augmented reality', 'Cybersecurity', 'Edge computing', and so on) has been used to validate the firms' degree of innovation and circumscribe the target to investigate. Following the selection process, 583 firms have been identified. 56% of the total number of firms are in the province of Pescara and 44% in the province of Chieti. More specifically, the observed firms are located in three main areas: the first includes an area ranging from the municipality of Pescara to that of Chieti, including neighboring municipalities. The second includes an area that goes from Ortona to the industrial zone of Val di Sangro, through the municipality of Lanciano. The third area, with a much smaller dimension, goes through the municipalities of Vasto and San Salvo (Fig 1).

Many firms are active in the 'Knowledge Intensive Activities', even called KIAs [25], which can be divided into manufacturing (science-based industries and specialized machinery and devices), services (software, consulting and engineering services, architecture and R&D) and art, culture and creative activities. Table 1 shows the frequency of the observed firms by Ateco 2007, which is the classification of economic activities adopted in Italy. Ateco 2007 classification is the national version of the European nomenclature, Nace Rev. 2. Ateco 2007 has been set out and approved by a Steering Committee that, in addition to Istat, is participated by the Ministries concerned, the Bodies which manage the main administrative data sources on firms (Agenzia delle Entrate, Chambers of Commerce, social security institutions) and the main business associations.

Since Ateco 2007 shows the same limitations as any SIC system, we propose an alternative approach to classify observed firms and industrial activities.

## Methodology

The proposed analysis uses algorithms of text mining and semantics that enable a streamlined processing of descriptive data. As mentioned, the reason for this massive text analysis lies in the need to tag in an informative and updated way the activities carried out by innovative firms, trying to capture their specializations and competencies.

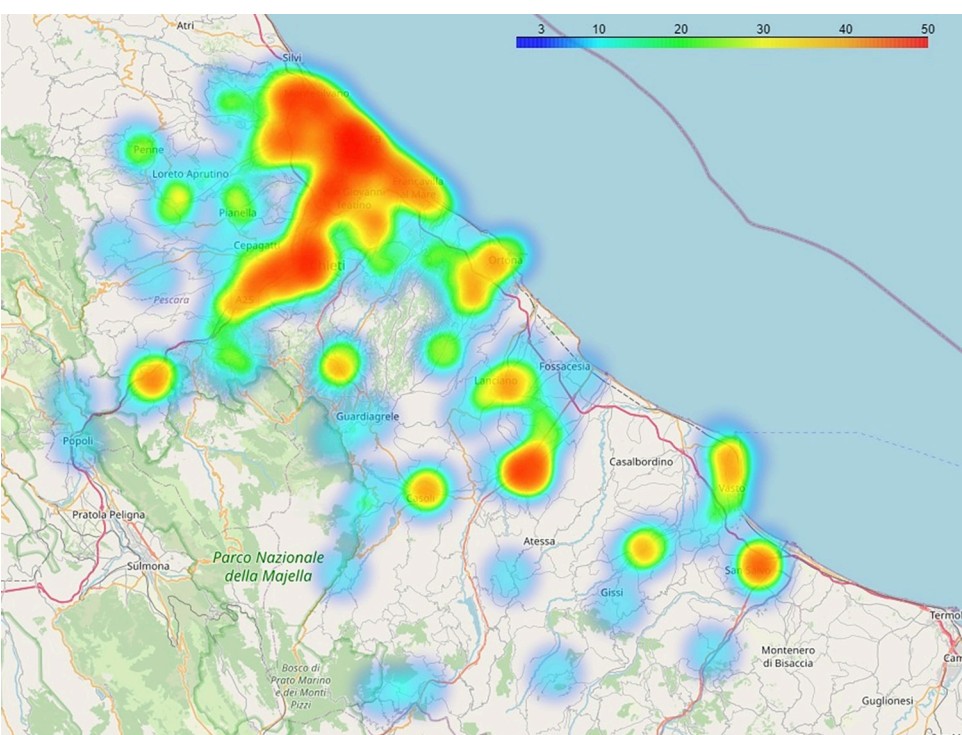

**Fig 1. The localization of the innovative firms in the provinces of Chieti and Pescara.** Heatmap based on the number of firms. Basemap: OpenStreetMap.

We processed multiple types of information sources such as corporate.html pages (through web crawling, where permitted) and.txt files on corporate purposes, together with other firms' descriptions if available. This has been the first step to obtaining a multilabel classification.

We made use of a 'general purpose' natural language recognition model based on machine learning algorithms pre-trained on different knowledge bases (such as Wordnet, Wikipedia, Dbpedia, and thousands of textbooks). We performed pre-processing procedures typical of text mining (e.g., lemmatization, stemming, stop-words, and so on). Additional modules have been used for spell-checking and language detection. We employed mixed models that draw on multiple existing and updated lists/ taxonomies and leverage access programming interfaces (APIs) to large libraries offered by software houses specializing in text analysis.

Afterwards, we switched to a 'specific model', calibrated to our research goal: what innovative firms do. We performed the labeling phase, that is assigning tags or labels to the observed firms, also by means of semantic understanding of the text. We carried out a first calibration based on some basic and easily understandable rules: for example, using corporate purposes to describe business activities requires the removal of standard ancillary activities (e.g., 'Ancillary to its principal business, the Company may also purchase, own, manage, use, update and develop, directly or indirectly, trademarks, patents and know-how concerning electronic tolling systems and related or connected activities').

We performed a semi-automatic check on tags to assess the quality of the generated output and added rules to reduce the noise acquired during the extraction phase. Given our interest in digital technologies, we carried out a further calibration focusing on ICT specializations. We proceeded with a normalization of the dataset. We used pre-defined algorithms to obtain a multi-label classification and assign each label to the categories or level of interest. We exploited taxonomies updated with technological evolution. We employed two families of

**Table 1. Innovative companies by Ateco 2007 two-digit class.**

| Ateco 2007 (2-digit class) | Frequency | Ateco 2007 (2-digit class) | Frequency |
|---|---|---|---|
| 06—extraction of crude petroleum and gas | 0.17% | 47—retail trade (except motor vehicles and motorcycles) | 0.34% |
| 09—mining support service activities | 1.89% | 49—land transport and transport via pipelines | 0.34% |
| 10 –manufacture of food products | 0.69% | 52—warehousing and support activities for transportation | 0.34% |
| 14—manufacture of wearing apparel | 0.51% | 53—postal and courier activities | 0.17% |
| 17—manufacture of paper and paper products | 0.51% | 58—publishing activities | 0.69% |
| 20—manufacture of chemicals and chemical products | 2.40% | 59 motion picture, video and television programme production, music and sound recording activities | 0.17% |
| 21—manufacture of basic pharmaceutical products and pharmaceutical preparations | 0.86% | 61—telecommunications | 1.54% |
| 22—manufacture of rubber and plastic products | 3.09% | 62—software production, IT consultancy and related activities | 22.81% |
| 23—manufacture of other non-metallic mineral products | 0.17% | 63—information service activities and other information services | 3.09% |
| 24 –manufacture of basic metals | 1.20% | 64—financial services activities (except insurance and pension funds) | 3.09% |
| 25—manufacture of fabricated metal products (except machinery and equipment) | 7.89% | 66—activities auxiliary to financial services and insurance activities | 0.34% |
| 26—manufacture of computers, electronic and optical products; electromedical equipment, measuring instruments and watches | 3.43% | 69—legal and accounting activities | 0.86% |
| 27—manufacture of electrical and non-electrical household equipment | 0.51% | 70 –activities of head offices and management consulting activities | 7.55% |
| 28—manufacture of machinery and equipment n.e.c. | 1.37% | 71—architectural and engineering activities; technical testing and analysis | 7.72% |
| 29—manufacture of motor vehicles, trailers and semi-trailers | 2.40% | 72—scientific research and development | 4.12% |
| 30—manufacture of other means of transport | 0.34% | 73—advertising and market research | 3.60% |
| 32—other manufacturing industries | 0.17% | 74—other professional, scientific and technical activities | 4.63% |
| 33—repair, maintenance and installation of machinery and equipment | 3.26% | 79—activities of travel agencies, tour operators and reservation services and related activities | 0.34% |
| 35—supply of electricity, gas, steam and air conditioning | 0.86% | 82—support activities for office functions and other business support services | 3.09% |
| 41—construction of buildings | 0.17% | 85—education | 1.20% |
| 42—civil engineering | 0.17% | 86—health care | 0.17% |
| 43—specialised construction work | 0.69% | 90—creative, artistic and entertainment activities | 0.17% |
| 46—wholesale trade (excluding motor vehicles and motorcycles) | 0.86% | | |

algorithms: extraction algorithms, which aim to identify the keywords characterizing the business activity, collected in a category called 'entities', and classification algorithms, which assign firms to pre-established categories. The former algorithms allow for the profiling of firms with specific details that firms themselves offer in the description of their own businesses. The classification algorithms create 'redundancy' by assigning firms to categories, which is fundamental to ensure the matching of firms.

Once extracted, after an accurate work of revision and standardization, the keywords are organized by category or level. A key aspect of this exercise concerns the choice of the taxonomies: these must be as broad and up to date as possible. We started with a set of taxonomies for classifying specializations and competencies using our previous knowledge base and external sources. The latter consist of expert-driven (where taxonomies are based on expert input) and data-driven classifications (where taxonomies are formulated using machine learning algorithms following the processing of large volumes of data).

For the first level (sectors) we referred to updated taxonomies adopted by open databases on innovative and high-tech sectors that collect hundreds of thousands of companies active in

**Table 2. List of sectors and relative frequency.**

| Sector | Frequency | Sector | Frequency |
|---|---|---|---|
| Information and Communication Technology | 13.03% | Automotive and transportation | 1.24% |
| Software | 10.83% | Chemistry | 1.24% |
| Consulting activities | 8.17% | Rubber, plastic, and non-metallic mineral processing | 1.24% |
| Internet & e-commerce | 7.84% | Finance and insurance | 1.20% |
| Plants and equipment | 6.85% | Biotechnology | 1.12% |
| Research | 5.93% | Construction | 1.12% |
| Manufacturing | 4.81% | Refined petroleum products | 0.91% |
| Professional, scientific, and technical activities | 4.61% | Entertainment, culture and sport | 0.71% |
| Electronics | 4.36% | Food and Beverage | 0.62% |
| Energy, environment and utilities | 4.36% | Retail | 0.54% |
| Hardware & Electrical Equipment | 3.15% | Agriculture | 0.33% |
| Human resources, training, and education | 2.99% | Pharmaceutical | 0.33% |
| Advertising | 2.61% | Textiles, clothing, tanning and footwear | 0.33% |
| Metallurgy & Metal Products | 2.12% | Mining activity | 0.25% |
| Industrial Machinery | 1.99% | Hotel and restaurant | 0.17% |
| Health and social care | 1.74% | Nanotechnology | 0.17% |
| Transport, logistics and storage | 1.62% | Rental, travel, and other business services | 0.08% |
| Other service activities | 1.33% | Non profit | 0.04% |

innovative sectors, including Crunchbase, Dealroom and AngelList. The list of sectors is provided in Table 2.

With regards to the other two categories (specializations and competencies), we relied on the one hand on taxonomies employed by software houses specializing in text analysis (such as, for example, Text Razor, Aylien, Dialogflow) and, on the other, on taxonomies that, although built in different contexts and for different purposes, are useful references. Among these last ones, we referred to vast documentation, including [26–28], and to existing taxonomies such as the Occupational Information Network [29], and the European Skills, Competences, Qualifications and Occupations [30]. The specialization category was created by exclusion: once the words were assigned to the categories of sectors and competencies (broadly defined), the more general or common words were attached to the category 'entities' (and, then, used to tag firms) or removed if not informative on firms' activities. After these steps, a large basket of words describing the activities carried out by the observed firms has been created. As known, competencies are an umbrella notion, very difficult to circumscribe [31]. In this paper, we define competence as the ability to apply knowledge and skills to achieve results, and for such a reason we opted to refer primarily to scientific disciplines to account for them. Therefore, within the sector in which the firms operate, it is possible to go further and characterize them based on the specializations that distinguish them within the industry on one side and on the disciplinary competencies possessed by the people working in the firms on the other. We assigned the 583 firms in target to more than a thousand categories: sectors (respectively, 32 unique words from corporate purposes and 36 from company websites), specializations (respectively, 310 unique words from corporate purposes and 931 unique words from websites), and competencies (respectively, 74 unique words from corporate purposes and 154 unique words from websites). Moreover, thousands of keywords in the category 'entities' were used to tag firms (respectively, 887 unique words from corporate purposes and 4614 unique words from websites).

Firms became strings of tags attached to different categories: sectors, specializations, competencies, and entities. Overall, more than 28 thousand keywords have been generated and

used in the database. Afterwards, it was possible to match firms to each other by means of common specializations (e.g., 'Supply chain management') and/ or competencies (e.g., 'Statistics').

In this way, we moved from working on textual data to relational data. Once the keywords have been created and sorted into categories, the analysis turns on the resulting matrices of adjacencies by which to bring close one firm to each other based on the number of categories and keywords that co-occur between them. Connecting firms through the co-occurrence of keywords means relying on well-known graph theory to investigate networks, nodes, and edges. As anticipated above, instead of focusing on networks of firms (583 nodes), too large to investigate in depth, we aggregate firms by sectors (36 nodes) approaching them based on firms' common specializations and/ or competencies. Nonetheless, also many other forms of aggregations are feasible as we will show. Before presenting the results, it is appropriate to point out some disadvantages of using network analysis. These disadvantages apply in general and a fortiori were encountered in our analysis. Firstly, the collection of textual data for network analysis requires careful and meticulous filtering and cleaning to guarantee the accuracy of the terms selected for the construction of the graphs. Secondly, network analysis does not always allow us to propose reliable comparisons between different graphs; in fact, often different networks represent phenomena with different structures and, therefore, are not comparable. Consequently, the analysis proposed below will focus on the most macroscopic evidence regarding the structure underlying the different graphs and the most significant patterns.

## Results

Instead of referring to the SIC system, we gathered information from companies' websites and corporate purposes and generated tags concerning firms' activities. Using such tags, we built a new dataset, which represented our starting point: the firms become sequences of tags that describe their activities or sectors, specializations, and competencies.

The resulting dataset provides a mapping of industrial activities with useful insights into firms' profile. The following sectors emerge as relevant: 'Information and communication' (13.03% of the total keywords assigned to the category sector), 'Software' (10.83%), 'Consulting activities' (8.17%), 'Internet & e-commerce' (7.84%), 'Plants and equipment' (6.85%), 'Research' (5.93%). Table 2 lists all sectors and their relative frequency.

**Table 3. Top 30 specializations and relative frequency.**

| Specialization | Frequency | Specialization | Frequency |
|---|---|---|---|
| Design | 3,36% | Computer networking | 0,87% |
| Digital Technology | 2,39% | Data transmission | 0,86% |
| Management | 2,29% | Online services | 0,86% |
| Information Technology management | 1,69% | Web software | 0,81% |
| Production and manufacturing | 1,59% | Computing | 0,80% |
| Application Software | 1,44% | Product management (marketing) | 0,78% |
| Data Management | 1,35% | Automation | 0,74% |
| Data | 1,24% | Web App | 0,73% |
| Software development | 1,20% | Business process | 0,72% |
| Product development | 1,19% | Networks | 0,72% |
| Analysis | 1,17% | Research and development | 0,71% |
| Infrastructure | 1,02% | Sustainable technologies | 0,70% |
| Materials production | 1,00% | Web 2.0 | 0,70% |
| Processing systems architecture | 0,92% | Business process management | 0,67% |
| Emerging Technologies | 0,89% | Digital media | 0,66% |

**Table 4. Top 30 competencies and relative frequency.**

| Competence | Frequency | Competence | Frequency |
|---|---|---|---|
| Engineering | 9,91% | Architecture | 1,69% |
| Computer Engineering | 6,83% | Chemistry (discipline) | 1,60% |
| Computer Science (discipline) | 5,86% | Industrial Engineering | 1,48% |
| Software engineering | 5,35% | Civil Engineering | 1,39% |
| Business administration | 4,11% | Scientific method | 1,21% |
| Communication | 3,67% | Environmental science | 1,21% |
| Mechanical Engineering | 3,52% | Strategic management | 1,18% |
| Marketing | 3,02% | Business law | 1,09% |
| Electrical Engineering | 2,93% | Materials science | 1,09% |
| Information Science | 2,90% | Health sciences | 1,09% |
| Electronic Engineering | 2,51% | Cybernetics | 0,95% |
| Systems Engineering | 2,04% | Project management | 0,92% |
| Construction Engineering | 2,04% | Corporate law | 0,86% |
| Science and Technology | 2,01% | Power Engineering | 0,83% |
| Telecommunications Engineering | 1,77% | Medicine | 0,83% |

The most common specializations are 'Design' (intended as the design phase of new products and/ or services, 3.36% of total keywords assigned to the category specialization), 'Digital Technology' (2.39%), 'Management' (2.29%), 'Information technology management' (1.69%), 'Production and manufacturing' (1.59%), 'Application software' (1.44%), 'Data management' (1.35%), 'Software development' (1.20%), 'Product development' (1.19%), 'Materials production' (1.00%), 'Online services' (0.86%), 'Web software'. (0.81%), 'Automation' (0.74%), 'Research and development' (0.71%), 'Sustainable technologies' (0.70%), and so on. Table 3 lists the top 30 specializations and their relevance in the database.

At the basis of the economic activities and firms' specializations there are people with competencies in specific disciplines. Among the main competencies found there are 'Engineering' (9.91% of the total keywords assigned to the category), 'Computer Engineering' (6.83%), 'Computer Science' (5,86%), 'Software Engineering' (5.35%), 'Business Administration' (4.11%), 'Communication' (3.67%), 'Mechanical Engineering' (3.52%), 'Marketing' (3.02%), 'Electrical Engineering' (2.93%), 'Information Science' (2.90%), 'Electronic Engineering' (2.51%), and so on. Table 4 lists the top 30 competencies and their relative frequency.

Co-occurrences of tags assigned to firms imply some proximity: two or more firms are close to each other if they are active in the same sector, are specialized in the same areas, and share some competencies.

As anticipated, most attempts directed to classify firms' activities aim also at capturing proximity between them. The most used measure uses the hierarchy of the SIC codes: the lower the class two firms share in the hierarchy, the more similar they are thought to be. According to this basic reasoning, firms in the same 5-digit class are more related than firms that only share the same 3-digit class. Fig 2 shows the network resulting from Ateco 2007, by which two firms are connected by a link of weight 5 if they share the same 5-digit class, by a link of weight 4 if they share the same 4-digit class, by a link of weight 3 if they share the same 3-digit class, by a link of weight 2 if they share the same 2-digit class. This network is undirected and weighted: this means that the higher the n-digit class, the heavier the edge between nodes.

The network of 583 firms is mostly fragmented, with several clusters apparently disconnected from each other. The average degree (that is, the average number of links per node) is

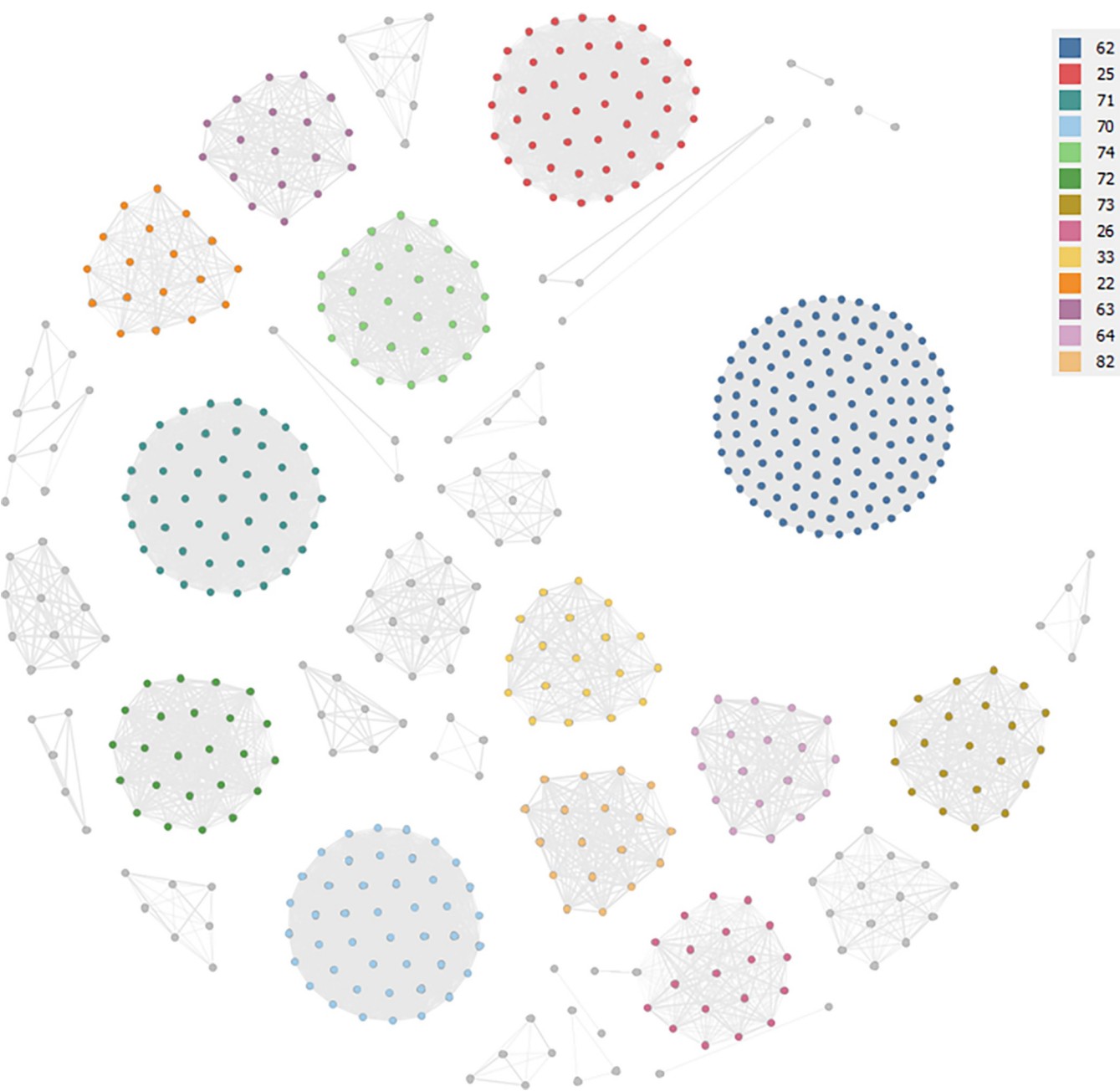

**Fig 2. The network of firms based on Ateco 2007.** Labels on larger (above 3% frequency) clusters of nodes indicate firms' Ateco 2007 2-digit class.

48.69, which would seem to indicate a rather high level of interaction between the nodes of the network but which, on closer examination, can be explained by the high level of interconnection between the nodes belonging to the software sector: 133 firms sharing the 5-digit class within economic activity '62 –Software'. Based on the value of the graph density (the ratio of the number of edges and the number of possible edges), it results in a rather limited level of interconnection, with a 0.09. High, as we can appreciate from the visualization, the level of dispersion of the network, with modularity (the measure of the strength of division of a network into groups or clusters; networks with high modularity have dense connections between the

nodes within groups but sparse connections between nodes in different clusters) that equals 0.60.

Is it correct to assume the presence of some links between firms because of the closeness within the Ateco 2007? In our opinion, the approximation is too crude: it is assumed a flow between two firms (only because) near in the SIC system, but nothing can be presumed on the possible content of the exchange. If isolating such content is a complex (if not impossible) task, a deeper investigation of firms' underlying industrial activities, specializations and competencies would allow for the deduction of something more about the emerging links.

The co-occurrence of tags across levels (sectors, specializations and competencies) and within the category 'entities' generates a highly interconnected network (Fig 3). This network, as well as all other following graphs, are undirected and weighted: the higher the number of co-occurrences, the heavier the edge between nodes.

The overall number of links is equal to 90972, which is 6,5 times larger than the previous one. The average degree is equivalent to 312.62, which suggests a high level of interconnection between the nodes. The value of the graph density is significantly higher than the previous graph, with a value of 0.54. Lower, as we can appreciate from Fig 3, the level of fragmentation of the network, with a modularity value equal to 0.20. While some clusters of firms are identifiable in terms of industry codes, such as the blue cluster (ATECO 62) at the bottom right, the green cluster (ATECO 72) above, and the red cluster (ATECO 25) at the bottom left, at the same time it is possible to discern the numerous mixtures between firms belonging to such ATECO codes and firms of different industries. For example, using text data ATECO 25 firms become close to ATECO 22 firms (orange nodes) and to other residual classes (nodes in grey, which refer to ATECO codes with a frequency of less than 3%). ATECO 72 firms at the top (in green) are close to ATECO 70 (light blue), ATECO 74 (light green) and ATECO 64 (violet) firms. The most relevant cluster in terms of nodes (blue, ATECO 62) is divided into three sub-clusters: the bottom right cluster (more colorful and wide-ranging in terms of ATECO 2-digit codes), the middle cluster with ATECO 73 firms (brown nodes) close to codes 72 and 74 (dark green and light green, respectively) and, finally, the bottom left cluster (mixed with ATECO 26 firms in bright pink).

Once we begin to aggregate firms by sector or specializations, we attain different and, often, more insightful views. Below, we provide some evidence that helps to understand which are the most central sectors, based on specializations and competencies respectively, and how close these are in Chieti and Pescara.

The first representation is useful to appreciate the interrelationships between firms, aggregated per sector of activity, to have a smaller number of nodes and a more intelligible graph. The links in the network are based on the co-occurrence of specializations between firms, aggregated by sector. The graph shows a rather high average degree (19.94), an average weighted degree (the average of sum of weights of the edges of nodes) of 9923.56, a network diameter (the maximum distance between any pair of nodes in the graph) of 3, and a graph density of 0.57. The average path length (the average number of steps along the shortest paths for all possible pairs of network nodes) is 1.43, while the modularity is 0.18 and the average clustering coefficient (the average degree to which nodes tend to cluster together) is 0.79 (Table 5).

Based on common specializations, the sectors 'Information and Communication technology' and 'Software' are at the center of the innovative economic system (Fig 4): looking at the weighted degree (that is, the sum of weights of the edges of nodes), the former shows 53958, while the latter 48092 (Table 6). Other sectors follow with lower values, such as 'Internet & e-commerce' (37620), 'Consulting activities' (25266) and 'Plants and equipment' (22038).

Even though we know that all these sectors can be considered interrelated based on similar specializations, we still aim at realizing 'how much' these sectors are close to each other. Such

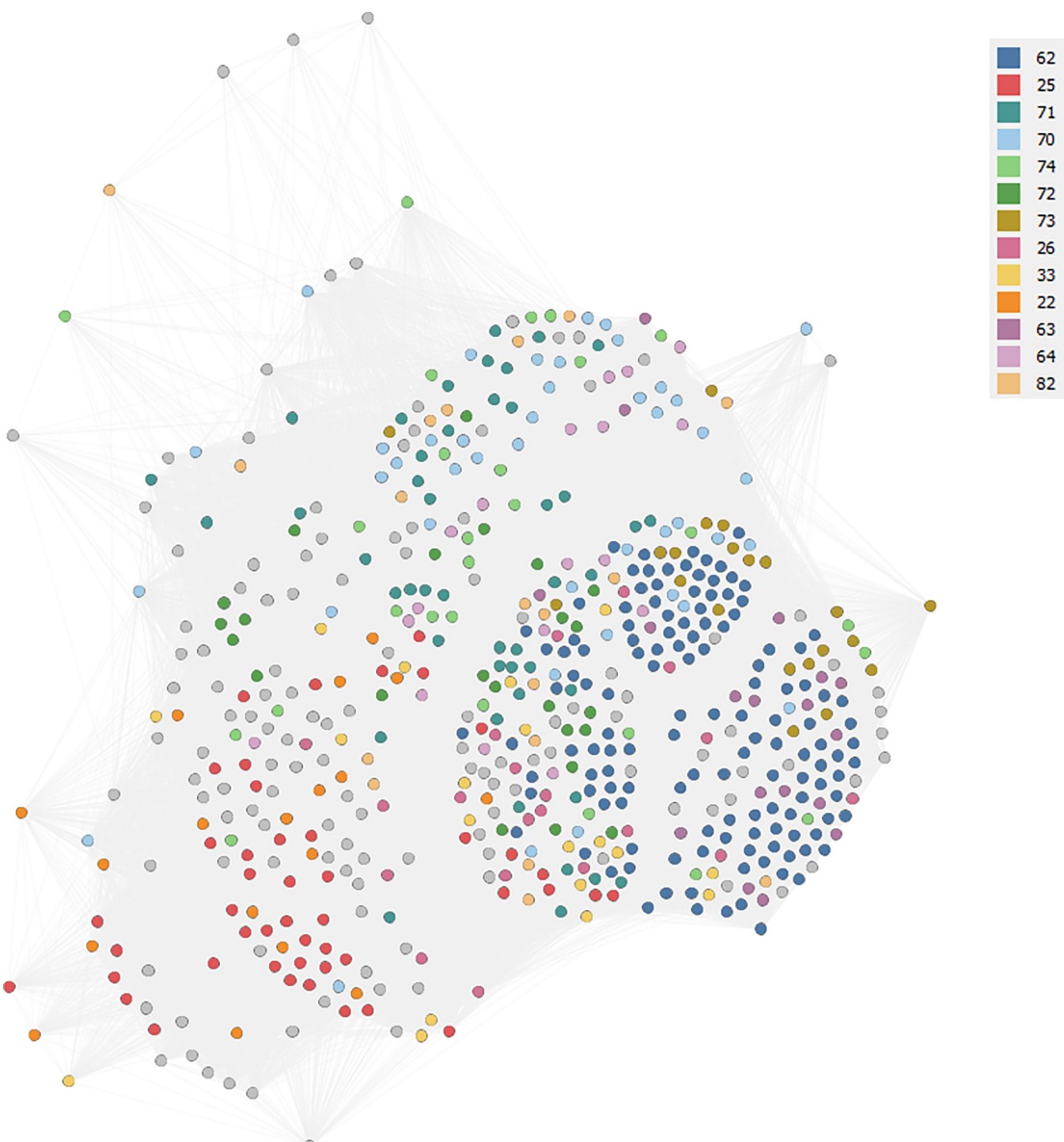

**Fig 3. A network based on co-occurrence of tags between firms in Chieti and Pescara.**

evidence becomes measurable through the metrics of the graphs. We report some measures of relevant relationships that can be used as benchmarks: the weight of the link between the firms active in 'Software' and those in 'Information and Communication technology' is equal to 12300, between firms in 'Information and Communication technology' and those in 'Internet

**Table 5. The metrics related to the network between sectors based on common specializations.**

| Metrics | Value/Coefficient |
|---|---|
| Average degree | 19.94 |
| Average weighted degree | 9923.56 |
| Network diameter | 3 |
| Average path length | 1.43 |
| Graph density | 0.57 |
| Modularity | 0.18 |
| Average clustering coefficient | 0.79 |

& e-commerce' is 12176, between 'Software' and 'Internet & e-commerce' is 9148, between 'Advertising' and 'Information and Communication technology' is 4254, 'Advertising' and 'Internet & e-commerce' is 4444, 'Electronics' and 'Plants and equipment' is 2298. Moreover, we can identify which are the specializations underlying the links between the above sectors, such as for example 'Digital technologies', 'Data management', 'Design of new products/ services', 'Software development' and 'Application software' just to mention the top 5 specializations.

Fig 5 shows a dense network of sectors based on the co-occurrence of competencies. This graph shows a rather high average degree (18.67), an average weighted degree of 2676.56, a

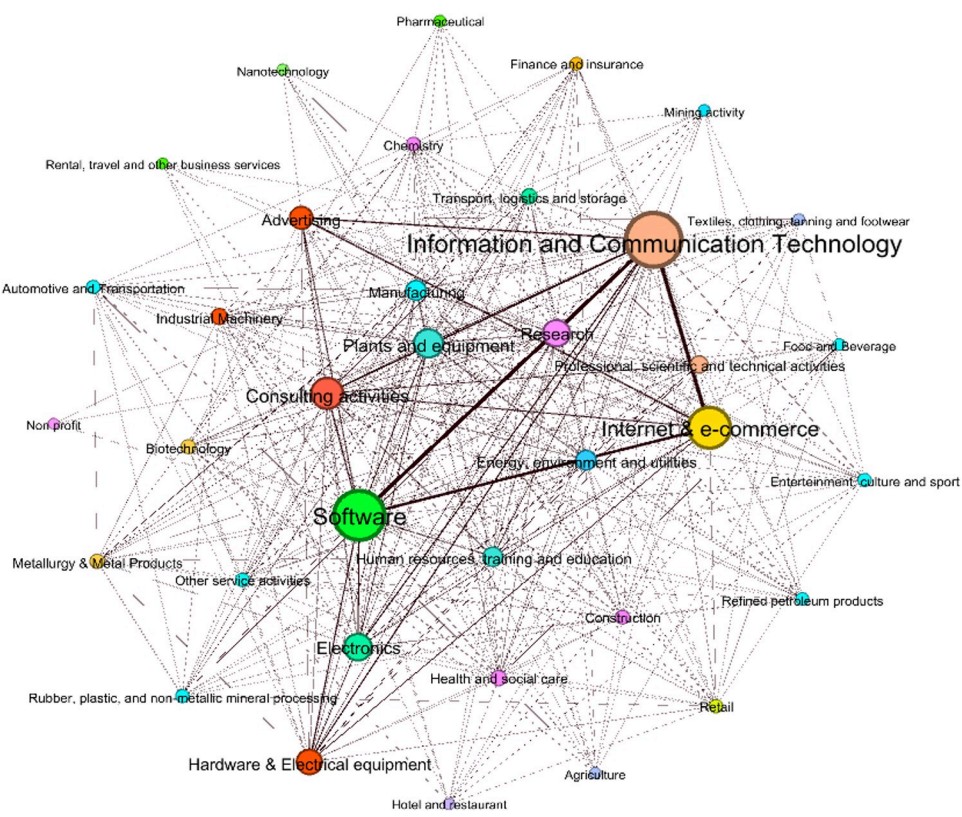

**Fig 4. The network between sectors based on common specializations.** The above sectors are well connected to other sectors such as, in primis, 'Electronics', 'Research', 'Hardware & electrical appliances', 'Advertising', and 'Manufacturing'.

**Table 6. The centrality of sectors based on the co-occurrence of specializations (weighted degree).**

| Nodes | Weighted Degree | Nodes | Weighted Degree |
|---|---|---|---|
| Information and Communication technology | 53958 | Construction | 3792 |
| Software | 48092 | Other service activities | 3504 |
| Internet & e-commerce | 37620 | Automotive and transport equipment | 3080 |
| Consulting activities | 25266 | Retail | 3054 |
| Plants and equipment | 22038 | Chemistry | 2984 |
| Electronics | 19792 | Rubber, plastic, and non-metallic mineral processing | 2904 |
| Research | 19506 | Entertainment, culture and sport | 2870 |
| Hardware & electrical appliances | 17396 | Finance and insurance | 2554 |
| Advertising | 14468 | Refined petroleum products | 2246 |
| Manufacturing | 11140 | Pharmaceutica | 1064 |
| Energy, environment and utilities | 11036 | Food and beverage | 976 |
| Human resources, training, and education | 10624 | Textiles, clothing, tanning and footwear | 976 |
| Professional, scientific, and technical activities | 7760 | Mining activity | 788 |
| Industrial machinery | 6500 | Agriculture | 580 |
| Health and social care | 5556 | Nanotechnologies | 532 |
| Transport, logistics and storage | 5466 | Hotel and restaurant | 454 |
| Metallurgy and metal products | 4232 | Rental, travel, and other business services | 144 |
| Biotechnology | 4200 | Non Profit | 96 |

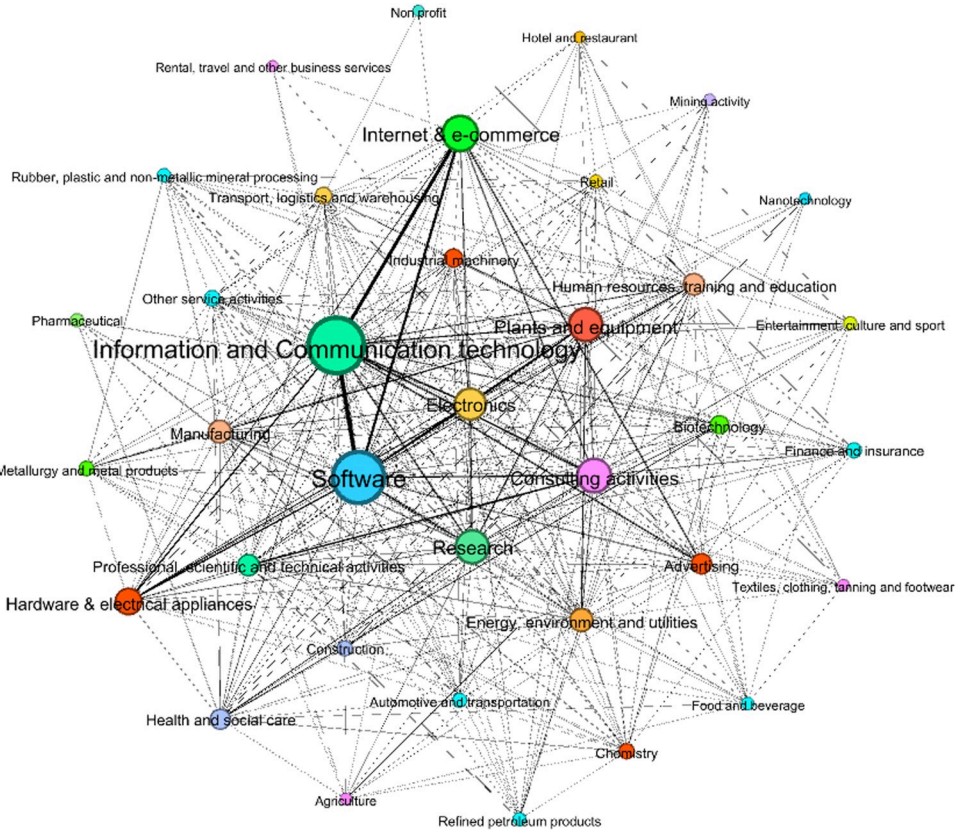

**Fig 5. The network between sectors based on common competencies.**

**Table 7. The metrics related to the network of sectors based on common competencies.**

| Metrics | Value/Coefficient |
|---|---|
| Average degree | 18.67 |
| Average weighted degree | 2676.56 |
| Network diameter | 3 |
| Average path length | 1.48 |
| Graph density | 0.53 |
| Modularity | 0.11 |
| Average clustering coefficient | 0.80 |

network diameter of 3, and a graph density of 0.53 (Table 7). It is interesting to notice that the competencies underlying the 'Research' and 'Consulting activities', together with ICT, 'Software' and 'Internet & e-commerce', work as *trait d'union* of the innovative activities.

In some respects, this is easy to understand since they are fundamental sectors, supporting the economic system (Table 8). Also, in this case, we can identify towards which sectors both 'Research' and 'Consulting activities' offer their specific knowledge. In this case, we detect the links between sectors based on the underlying competencies.

The main target sectors are 'Energy, environment and utilities', 'Plants and equipment', 'Professional, scientific, and technical activities' and 'Transport, logistics and storage'. We can focus on some links that can be taken as reference: the weight of the link between the firms active in 'Software' and those active in 'Information and Communication technology' is equal to 3450, between 'Information and Communication technology' and 'Internet & e-commerce' equals 2720, 'Software' and 'Electronics' is 1270, between 'Consulting' and 'Information and Communication technology' is 1248, between 'Consulting' and 'Professional, scientific and technical activities' is 1176, between 'Plants and equipment' and 'Software' is 1024, between 'Research' and 'Information and Communication technology' is 972.

**Table 8. The centrality of sectors based on the co-occurrence of competencies (weighted degree).**

| Nodes | Weighted Degree | Nodes | Weighted Degree |
|---|---|---|---|
| Information and Communication technology | 16090 | Chemistry | 1508 |
| Software | 14546 | Construction | 1466 |
| Internet & e-commerce | 8660 | Finance and insurance | 1430 |
| Consulting activites | 8168 | Metallurgy and metal products | 1300 |
| Plants and equipment | 8016 | Automotive and means of transport | 1104 |
| Research | 7930 | Rubber, plastic, and non-metallic mineral processing | 872 |
| Electronics | 7254 | Pharmaceutical | 780 |
| Hardware & electrical appliances | 5444 | Entertainment, culture and sport | 772 |
| Energy, environment and utilities | 4182 | Refined petroleum products | 760 |
| Manufacturing | 4086 | Retail | 702 |
| Human resources, training, and education | 3700 | Food and beverage | 470 |
| Professional, scientific, and technical activities | 3620 | Agriculture | 248 |
| Advertising | 3260 | Nanotechnology | 230 |
| Health and social care | 3192 | Textiles, clothing, tanning and footwear | 198 |
| Biotechnology | 2542 | Mining activity | 150 |
| Industrial machinery | 2492 | Hotel and restaurant | 128 |
| Transport, logistics and storage | 1966 | Rental, travel, and other business services | 60 |
| Other service activities | 1638 | Non profit | 40 |

**Table 9. The metrics related to the network of specializations based on common competencies.**

| Metrics | Value/Coefficient |
| --- | --- |
| Average degree | 15.27 |
| Average weighted degree | 6207.48 |
| Network diameter | 6 |
| Average path length | 2.28 |
| Graph density | 0.12 |
| Modularity | 0.22 |
| Average clustering coefficient | 0.80 |

Even in this case, we are not only able to measure the weight of the links, but also identify the driving competencies behind such links: 'Computer engineering', 'Mechanical engineering', 'Chemistry' and 'Business disciplines'.

The reasoning above is replicable to a network of specializations. For visualization purposes, we are restricting the network to the top 130 specializations (Table 9). In this case, specializations are brought close to each other if these are accompanied in one or more firms by the same competencies: the higher the number of firms sharing the same specializations and competencies, the heavier the edge between specializations in the graph.

Data management and digital technologies play a pivotal role in terms of the underlying specific competencies in Chieti and Pescara (Fig 6).

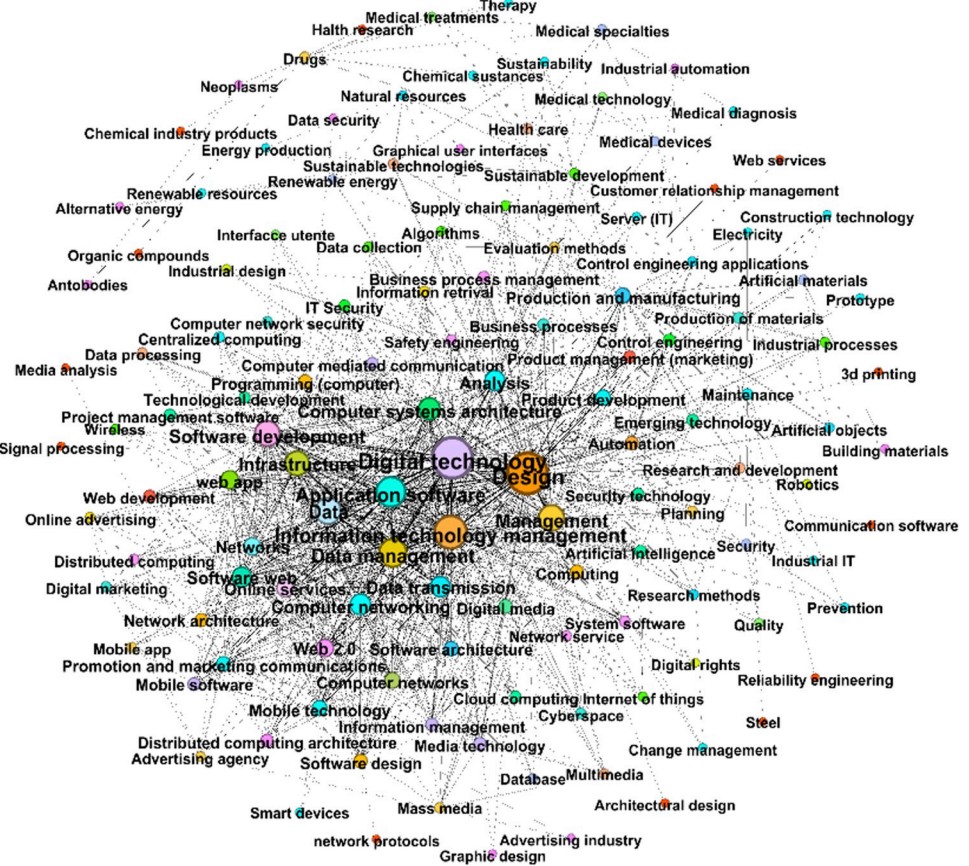

**Fig 6. The network between specializations based on common competencies.**

**Table 10. Top 30 specializations based on the co-occurrence of competencies (weighted degree).**

| Nodes | Weighted Degree | Nodes | Weighted Degree |
|---|---|---|---|
| Design of new products/ services | 50190 | Web 2.0 | 14388 |
| Digital Technology | 50146 | Networks | 14070 |
| Information Technology Management | 37812 | Online services | 13896 |
| Application software | 32978 | Computer Networks | 11764 |
| Data management | 30302 | Production and manufacturing | 10936 |
| Data | 27412 | Mobile technology | 10800 |
| Management | 27084 | Product Development | 10644 |
| Software Development | 26472 | Software architecture | 9862 |
| Infrastructure | 24496 | Promotion and marketing communications | 9568 |
| Architecture of processing systems | 21036 | Digital media | 8546 |
| Computer networking | 19242 | Software design | 8452 |
| Data transmission | 18738 | Computing | 8134 |
| Analysis | 18604 | Automation | 8026 |
| Web software | 17936 | Programming (computers) | 7918 |
| Web app | 14968 | Network architecture | 7532 |

Among the most central and interconnected specializations in the network, there are 'Information Technology Management', 'Data', 'Data transmission', 'Mobile technology', 'Digital media', 'Programming' and 'Automation', which have at their base competencies in engineering disciplines such as, in order of relevance, 'Electrical engineering', 'Computer science', 'Electronic engineering', 'Mechanical engineering' and 'Telecommunications engineering' (Table 10).

The strongest links are between firms specialized in 'Design' and those in 'Digital technology', which weighs 2246, between specializations 'Information Technology Management' and 'Digital technology' (2016), 'Information Technology Management' and 'Data management' (1386), 'Data' and 'Digital technology' (1356), 'Digital media' and 'Digital technology' (724).

## Conclusions

The paper intended to propose an original method to tag innovative firms and classify industrial activities. Instead of referring to SIC codes, we gathered information from companies' websites and corporate purposes, extracted keywords and generated tags concerning firms' activities. Therefore, firms became sequences of tags ordered on different levels or categories: sectors, specializations, and competencies.

Why transform firms' descriptive texts into keywords? There are at least two reasons for this. The first lies in the fact that investigating innovative activities is a complicated task, especially in modern times when technological evolution is rapid, and innovation is incessant. Therefore, starting from a large and updated information base, even though fluid and dynamic, helps a lot. The second lies in the fact that the keywords can be used to link firms and sectors (groups of firms) based on the researcher's interest, for example using underlying specializations or competencies. Also, as seen, firms' specializations can be grouped based on underlying competencies.

Our paper used text mining and semantic algorithms to tag innovative firms and offer an alternative perspective to classify industrial activities. Evidence is interesting because allows us to understand what firms do in a more penetrating and updated way than by referring to SIC codes. Keywords are generated from firms' descriptive texts and, for this reason, are more informative than short and static industrial classifications. Moreover, through matching firms'

keywords, we were able to explore the degree of interconnection between firms, a measure by which researchers can derive industrial proximity. We were able to bring close firms based on the tags they have in common (e.g., linking all firms that are specialized in 'supply chain management'). Similarly, we linked firms because of the specific competencies on which they build their own specializations (e.g., connecting firms that share 'computer engineering'). In general, as well known, such exercises are useful because they allow circumscribing of specific clusters of economic activities, from which firms absorb knowledge and ideas and in which they evolve and transform.

Using the keywords assigned to firms helps to discover a world of existing and intangible relationships, often hidden, which represent in some ways the DNA of the economic system under observation and allow to capture the 'energy' behind new entrepreneurial initiatives, innovative projects, interindustry collaborations, and synergic partnerships. Taking inspiration from a metaphor proposed by [32] such an investigation of the economic activities seems to recall the activity of biologists when they look at phenotypes (the physical and functional characteristics of an organism) as expressions of genotypes (the information embodied in the DNA of an organism).

As illustrated above, in Chieti and Pescara the sectors, ICT and 'Software' are at the center of the innovative economic system by looking at the specializations. By using competencies, it was interesting to notice how 'Research' and 'Consulting activities', together with ICT, 'Software' and 'Internet & e-commerce', work as *trait d'union*. Below the link between firms' activities and specializations, there are plenty of competencies such as 'Computer engineering', 'Chemistry', 'Business disciplines', 'Electrical engineering', 'Computer science', 'Electronic engineering', 'Mechanical engineering' and 'Telecommunications engineering'.

These are just a few of the insights that can emerge from the proposed method: the results identified by the researcher will be the more precise, the greater the desire the researcher has in going into the detail of a specific industrial area. The proposed analysis shows some limitations. First, the performance of the exercise depends on the quality of the text data sources. This implies two orders of consideration. The former relates to the choice of the source data and concerns, as mentioned above, the wide divergence existing between websites and corporate purposes. The latter regards the variability that can exist in terms of the broadness of the descriptions of business activities between one firm and another: there are firms that show very precise descriptions of their business and firms that, instead, tend to describe themselves in an oversimplified way. Secondly, the selection of the firms to investigate might be biased by the scarce information provided by companies' websites and/ or corporate purposes. This is crucial in those cases in which the researcher is interested in defining a precise target of firms. Thirdly, there is the question of the taxonomy employed. This is never complete and often is the result of a combination of several taxonomies, as illustrated above, each characterized by some limitations in breadth and/ or in depth. In this sense, the diffusion of more empirical exercises aimed at describing firms' activities, as well as the introduction of new databases and/ or the enrichment of existing libraries, can contribute to making the classification a more robust and less questionable step.

The proposed exercise suffers from a major limitation. SIC codes are useful because they classify firms over years, going back in time, while the text data approach would not allow for the identification of a firm's sector for each year in the last five or 15 years, unless the firms' descriptions have been captured over time and the information archived, keeping the extraction methodology and the linking taxonomy intact.

One point remains robust: the breadth and depth of the information processed and the chance of building numerous graphs to relate firms (and groups of firms) in a very precise way. The analysis provides policymakers with a detailed and comprehensive picture of the

innovative trajectories underlying the industrial structure. Some of the questions that the above results help answer are: Which firms (are similar or) have a specific specialization or competence? How common is that specialization or competence in the observed group of firms? What competencies link firms active in apparently distant industries?

## Author Contributions

**Conceptualization:** Alessandro Marra, Cristiano Baldassari.

**Data curation:** Cristiano Baldassari.

**Formal analysis:** Alessandro Marra.

**Investigation:** Alessandro Marra.

**Methodology:** Alessandro Marra, Cristiano Baldassari.

**Resources:** Cristiano Baldassari.

**Software:** Cristiano Baldassari.

**Supervision:** Alessandro Marra.

**Validation:** Cristiano Baldassari.

**Visualization:** Cristiano Baldassari.

**Writing – original draft:** Alessandro Marra.

**Writing – review & editing:** Alessandro Marra.

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
