## [Decision Letter · Decision Letter 0]

14 Dec 2021

PONE-D-21-29097Using text data (instead of SIC codes) to tag innovative firms and measure proximityPLOS ONE

Dear Dr. Marra,

Thank you for submitting your manuscript to PLOS ONE. After careful consideration, we feel that it has merit but does not fully meet PLOS ONE’s publication criteria as it currently stands. Therefore, we invite you to submit a revised version of the manuscript that addresses the points raised during the review process.

Overall, while there is a contributing point in the paper, there is mismatch between the current literature review, title, discussion and contributing area. The title should be revised as well. I recommend authors to align well the introduction, literature review, analysis and conclusion.

We look forward to receiving your revised manuscript.

Kind regards,

Wonjoon Kim, Ph.D

Academic Editor

PLOS ONE

“*The authors gratefully acknowledge financial support from Fondirigenti G. Taliercio (reference CIG: 8188368708). The funding source had no involvement in study design, in the collection, analysis and interpretation of data, in the writing of the paper, and in the decision to submit the article for publication. The usual disclaimer applies”*

 “The author (Alessandro MARRA) gratefully acknowledges financial support from Fondirigenti G. Taliercio (reference CIG: 8188368708). The funding source had no involvement in study design, in the collection, analysis and interpretation of data, in the writing of the paper, and in the decision to submit the article for publication. The usual disclaimer applies”

“NO authors have competing interests”

5. We note that [Figure 1] in your submission contain [map/satellite] images which may be copyrighted. All PLOS content is published under the Creative Commons Attribution License (CC BY 4.0), which means that the manuscript, images, and Supporting Information files will be freely available online, and any third party is permitted to access, download, copy, distribute, and use these materials in any way, even commercially, with proper attribution. For these reasons, we cannot publish previously copyrighted maps or satellite images created using proprietary data, such as Google software (Google Maps, Street View, and Earth). For more information, see our copyright guidelines: http://journals.plos.org/plosone/s/licenses-and-copyright.

Reviewers' comments:

Reviewer's Responses to Questions

**Comments to the Author**

1. Is the manuscript technically sound, and do the data support the conclusions?

Reviewer #1: Yes

Reviewer #2: Partly

2. Has the statistical analysis been performed appropriately and rigorously? 

Reviewer #1: N/A

Reviewer #2: N/A

3. Have the authors made all data underlying the findings in their manuscript fully available?

Reviewer #1: Yes

Reviewer #2: No

4. Is the manuscript presented in an intelligible fashion and written in standard English?

Reviewer #1: Yes

Reviewer #2: Yes

5. Review Comments to the Author

Reviewer #1: This study is more of an exploratory study than a general research paper that reveals causal relationships with research questions. It would be better to express this in the title. It is recommended to mention the general shortcomings and uncertainties of network analysis in relation to this study.

Reviewer #2: This paper deals with an interesting topic, which is calculation proximity not based on SIC code but based on text mining of firms’ webpage (according to their abstract). Authors used two data sets, which are start-up and SME DB from Italian Chamber of Commerce, and text data that they crawled from the firm’s webpage. Then, they firstly did the multilabel classification and assign the labels. After several revision steps, they got the organized classification including categories and levels. They chose the taxonomy, which is consisted with specialization and competence. Finally, they classified 583 firms into 32 sectors, 310 specializations and 74 competences. At the result part, they mostly presented the summary of their new classification by visualizing results and showing tables.

Their contribution is that authors suggest novel method of industrial classification based on text mining that can be better performed (in what aspect?) comparing with Ateco 2007 or general SIC codes.

Overall, although this research suggests the new way of industrial classification, they seek their contribution from the literature on proximity. This mismatch between what they do and what they want to do may mislead readers and make the article lose its coherency. The title of the article heads toward the research stream on proximity, while their literature review and other parts are more likely heading to the industrial classification. I assume that authors’ definition on the industrial classification is sorely based on the proximity among firms. However, there are several rationales behind the classical method of industrial classification. Clustering firms based on what they do, which can be captured the proximity, can be one of the rationales. To solve this mismatch, one option for authors can be to connect the literature on the industrial classification and that on the proximity. The other option might be focusing only on the literature on the industrial classification, since the literature on the proximity pursues beyond the classification of industry and they mostly use the proximity measure to explore the chance of economic development in regions/countries. Therefore, since this article is focusing on the classification itself, it would be better to focus on and look for their contribution from the literature on the industrial classification without mentioning proximity measure. If so, author also need to compare the classical way of industrial classification (i.e., SIC) and their text data-driven method of industrial classification to address their contribution to the literature.

Minor comments:

1. They are extracting the list of firms for text mining from the start-ups and SME DB, which Italian Chambers of Commerce provides. This DB seems to cover all area of Italy. Then why author only choose some region instead of entire looking at the entire Italy?

2. Authors insist that they are looking at “innovative” firms by using keywords, such as digital technology, artificial intelligence, industrial automation, robotics and so on. For me, looking at the entire firms without narrowing their sample sounds more interesting. In addition, authors find that the innovative center is at the sectors “Information and Communication technology” and “Software” based on common specializations. Because they already filtered the keyword, this result is guaranteed from the beginning. If they cover all the firms without filtering using the keywords, their result becomes more meaningful.

3. In the methodology part, authors choose two taxonomies, which are specialization and competences. Considering that they present their results based on these two taxonomies, more explanation what these two means is required. Please add the explanation and make a reader understood the meaning of the visualized result using the two taxonomies.

4. Figures are tables are not self-complete. Please add more explanation of figures and tables to its caption. For example, what do the colors represent for in figure 2.

5. Overall, figures are not reader friendly. For example, hairball like figure 4 should be revised. Please use the way of network visualization in Hidalgo et al. (2009). You can find the supplementary material that describe their visualization method from here: https://www.science.org/doi/10.1126/science.1144581. Or you can follow the visualization method following Gao et al. (2021). Please see the supporting material of this research as well.

6. The last paragraph of the literature review just put the list of research articles. Please explain the meaning of each literature to utilize it to address your contribution.

7. Please add page numbers.

Reference:

Hidalgo, César A., Bailey Klinger, A-L. Barabási, and Ricardo Hausmann. "The product space conditions the development of nations." Science 317, no. 5837 (2007): 482-487.

Gao, Jian, Bogang Jun, Alex ‘Sandy Pentland, Tao Zhou, and César A. Hidalgo. "Spillovers across industries and regions in China’s regional economic diversification." Regional Studies (2021): 1-16.

6. PLOS authors have the option to publish the peer review history of their article (what does this mean?). If published, this will include your full peer review and any attached files.

Reviewer #1: No

Reviewer #2: No

---

## [Author Response · Author response to Decision Letter 0]

28 Jan 2022

Through the Response to reviewers (see other attached file) we intend to provide the Editor and the referees with a detailed list of the revisions brought to the manuscript. We considered all the editor and referees’ comments. 

At the end of the process of revision, our paper was substantially improved in several respects. Thank you for your time and feedback.

In the Response to reviewers, the editor and referees’ suggestions have been collected and organized according to the order followed in the mail sent to the authors (PLOS ONE Decision: Revision required [PONE-D-21-29097] - [EMID:36dfc19c49cd2b78]).

---

## [Decision Letter · Decision Letter 1]

3 Jun 2022

Using text data (instead of SIC codes) to tag innovative firms and classify industrial activities

PONE-D-21-29097R1

Dear Dr. Marra,

We’re pleased to inform you that your manuscript has been judged scientifically suitable for publication and will be formally accepted for publication once it meets all outstanding technical requirements.

Kind regards,

Wonjoon Kim, Ph.D

Academic Editor

PLOS ONE

Additional Editor Comments (optional):

Please submit the final version reflecting the reviewer 2's minor comment.

Reviewers' comments:

Reviewer's Responses to Questions

**Comments to the Author**

1. If the authors have adequately addressed your comments raised in a previous round of review and you feel that this manuscript is now acceptable for publication, you may indicate that here to bypass the “Comments to the Author” section, enter your conflict of interest statement in the “Confidential to Editor” section, and submit your "Accept" recommendation.

Reviewer #1: All comments have been addressed

Reviewer #2: All comments have been addressed

2. Is the manuscript technically sound, and do the data support the conclusions?

Reviewer #1: Yes

Reviewer #2: Yes

3. Has the statistical analysis been performed appropriately and rigorously? 

Reviewer #1: Yes

Reviewer #2: Yes

4. Have the authors made all data underlying the findings in their manuscript fully available?

Reviewer #1: Yes

Reviewer #2: No

5. Is the manuscript presented in an intelligible fashion and written in standard English?

Reviewer #1: Yes

Reviewer #2: Yes

6. Review Comments to the Author

Reviewer #1: All comments have been addressed, and the paper was improved considerably.

Measuring and analyzing innovative phenomena is very difficult, and it has long been an ongoing research topic for researchers in this field. In that sense, this study is meaningful in that it suggested a new method of measuring innovation. I hope various applications of this approach uses will be made through future researches.

Reviewer #2: Most of my previous comments are well addressed. However, there are some minor comments on grammar and title. Please remove parentheses from the title. Also, I can come across some grammatical error, while reading it. Please check grammar. Lastly, please write the name of classification in 2 digit instead of the class number in Table 1.

7. PLOS authors have the option to publish the peer review history of their article (what does this mean?). If published, this will include your full peer review and any attached files.

Reviewer #1: No

Reviewer #2: No

---

## [Editor Report · Acceptance letter]

10 Jun 2022

PONE-D-21-29097R1 

Using text data instead of SIC codes to tag innovative firms and classify industrial activities 

Dear Dr. Marra:

I'm pleased to inform you that your manuscript has been deemed suitable for publication in PLOS ONE. Congratulations! Your manuscript is now with our production department. 

Kind regards, 

on behalf of

Dr. Wonjoon Kim 

Academic Editor

PLOS ONE